# Universal Domain Adaptation for Robust Handling of Distributional Shifts in NLP

**Hyuhng Joon Kim[1], Hyunsoo Cho[1,2], Sang-Woo Lee[2,3,4], Junyeob Kim[1],**
**Choonghyun Park[1], Sang-goo Lee[1,5], Kang Min Yoo[1,2,3*], Taeuk Kim[6*]**

[1]Seoul National University, [2]NAVER Cloud, [3]NAVER AI Lab,
[4]KAIST, [5]IntelliSys, [6]Hanyang University

{heyjoonkim,johyunsoo,juny116,pch330,sglee}@europa.snu.ac.kr
{sang.woo.lee,kangmin.yoo}@navercorp.com
kimtaeuk@hanyang.ac.kr

## Abstract

When deploying machine learning systems to the wild, it is highly desirable for them to effectively leverage prior knowledge to the unfamiliar domain while also firing alarms to anomalous inputs. In order to address these requirements, Universal Domain Adaptation (UniDA) has emerged as a novel research area in computer vision, focusing on achieving both adaptation ability and robustness (i.e., the ability to detect out-of-distribution samples). While UniDA has led significant progress in computer vision, its application on language input still needs to be explored despite its feasibility. In this paper, we propose a comprehensive benchmark for natural language that offers thorough viewpoints of the model's generalizability and robustness. Our benchmark encompasses multiple datasets with varying difficulty levels and characteristics, including temporal shifts and diverse domains. On top of our testbed, we validate existing UniDA methods from computer vision and state-of-the-art domain adaptation techniques from NLP literature, yielding valuable findings: We observe that UniDA methods originally designed for image input can be effectively transferred to the natural language domain while also underscoring the effect of adaptation difficulty in determining the model's performance.

## 1 Introduction

Deep learning models demonstrate satisfactory performance when tested on data from the training distribution. However, real-world inputs encounter novel data ceaselessly that deviate from the trained distribution, commonly known as distributional shift. When confronted with such inputs, machine learning models frequently struggle to differentiate them from regular input. Consequently, they face challenges in adapting their previously acquired knowledge to the new data distribution, resulting

---

*Co-corresponding authors.

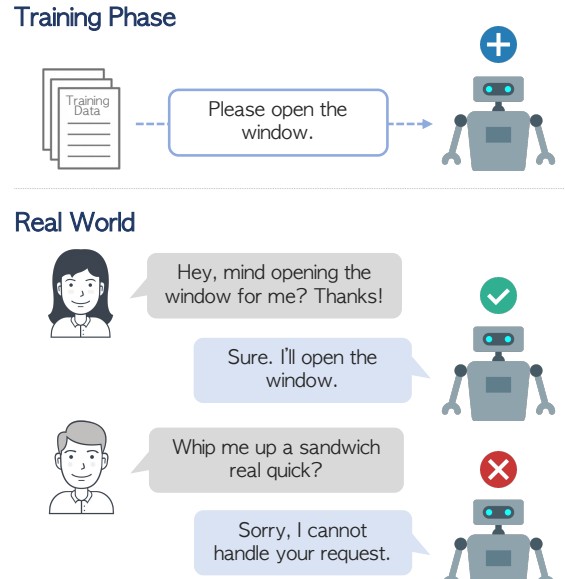

Figure 1: The model trained with formal language (source domain) will likely face spoken language (target domain) in the real world. The model is expected to properly handle such transferable input despite the distributional shift. (middle) At the same time, the model should discern unprocessable inputs (bottom) from the target domain.

in a significant drop in performance (Ribeiro et al., 2020; Miller et al., 2020; Hendrycks et al., 2020). The aforementioned phenomenon represents a long-standing challenge within the machine learning community, wherein even recent cutting-edge language models (OpenAI, 2023; Touvron et al., 2023; Chowdhery et al., 2022; Brown et al., 2020) do not serve as an exception to this predicament (Wang et al., 2023).

In response to these challenges, existing literature proposes two distinct approaches. The first approach, known as Domain Adaptation (DA) (Blitzer et al., 2006; Ganin et al., 2016b; Karouzos et al., 2021; Wu and Shi, 2022), endeavors to establish alignment between a new set of data from an unknown distribution and the model's prior knowl-

edge distribution. The objective is to enhance the model's generalization capability and reduce performance drop springing from the distributional shift. In parallel, a distinct line of work, referred to as out-of-distribution (OOD) detection (Aggarwal, 2017; Hendrycks and Gimpel, 2017; Hendrycks et al., 2019; Cho et al., 2021), focuses on discerning inputs originating from dissimilar distributions. They opt to circumvent potential risks or disruptions arising from shifted inputs, thereby enriching system robustness and resilience.

While both approaches offer unique advantages addressing specific distributional shifts, integrating their merits could substantially enhance robustness. In pursuit of this objective, a novel field called **Universal Domain Adaptation (UniDA)** (You et al., 2019) has emerged, aiming to harness the synergies of both OOD detection and DA when confronted with distributional shifts. UniDA leverages the best of the two worlds and offers comprehensive perspectives that integrate the merits of these two research areas. The essence of UniDA lies in measuring the uncertainty of the data from the shifted distribution precisely. Then, we can enhance the model's transferability by distinguishing portions of low-uncertainty inputs that can be adequately handled with the current model's knowledge. Simultaneously, we enrich the robustness of the model to OOD inputs by discerning the remaining samples that cannot be processed normally. However, distinguishing between these inputs and properly processing them becomes increasingly challenging without explicit supervision.

Despite the versatility of UniDA, this topic has yet to be explored in the Natural Language Processing (NLP) literature. As a cornerstone in enhancing reliability against distributional shifts in NLP, we introduce a testbed for evaluating the model's robustness in a holistic view. First, we construct various adaptation scenarios in NLP, utilizing an array of thoughtfully selected datasets. To discern the degree to which our proposed datasets incorporate the various degree of challenges in UniDA, we define two novel metrics: **Performance Drop Rate (PDR)** and **Distinction Difficulty Score (DDS)**. Using these metrics, We verify that our testbed captures a broad spectrum of distributional shifts. Finally, based on the suggested setting, we systematically compare several UniDA methods inherently designed for the task, against heuristic combinations of previous approaches for the parts

of the problem, i.e., OOD detection and DA.

Our empirical results show that UniDA methods are fully transferable in the NLP domain and can robustly respond to various degrees of shift. Moreover, we find out that the adaptation difficulty notably affects the performance of the methods. In certain circumstances, DA methods display comparable or even better performance. We release our dataset, encouraging future research on UniDA in NLP to foster the development of more resilient and domain-specific strategies.[2]

## 2 Universal Domain Adaptation

### 2.1 Problem Formulation

Distributional shift refers to a situation where the joint distribution $P$ estimated from the training dataset fails to adequately represent the wide range of diverse and complex test inputs. More formally, a distributional shift arises when the test input $x_{\text{test}}$ originates from a distant distribution $Q$, which is not effectively encompassed by the current trained distribution $P$.

One of the most prevalent research areas addressing this distributional shift includes OOD detection and DA. OOD detection aims to strictly detect all inputs from $Q$ to enhance the model's reliability. Although distribution $Q$ demonstrates a discernibly different distribution from the distribution $P$, the trained model can still transfer a subset of instances from $Q$, overcoming the inherent discrepancy between $P$ and $Q$. This particular capability serves as a fundamental motivation underlying the pursuit of DA. UniDA endeavours to integrate the merits of both fields, thereby enhancing both the generalizability and reliability of the model. Specifically, let us divide the target distribution $Q$ into disjoint subsets $H$, which share the same label space with source distribution $P$ and its complement $I$ ($Q = H \cup I$). The objective of UniDA is to enrich the robustness of the model by flexibly transferring existing knowledge to *transferable* samples from $H$ while firing alarms to *unknown* samples from $I$.

### 2.2 Challenges in UniDA

UniDA models should be capable of accurately capturing the underlying reasons behind the shift, thereby enabling the discrimination between *transferable* samples and *unknown* samples. Among the diverse categories of causes, the **domain gap**

---

[2]The dataset is available at https://github.com/heyjoonkim/universal_domain_adaptation_for_nlp

and the **category gap** (You et al., 2019) emerge as pivotal factors, each exerting a substantial impact on the overall complexity of the UniDA problem. While these concepts have previously been defined in a rather vague manner, we deduced the necessity for a more explicit definition. Thus, we set forth to redefine the concepts of domain and category gap more explicitly.

Domain gap refers to the performance drop when a model trained on $P$ fails to correctly process *transferable* inputs due to the fundamental discrepancy between $P$ and $H$, i.e., a domain shift. A dataset with a higher domain gap amplifies the problem's difficulty as the trained model becomes more susceptible to misaligning *transferable* samples.

A category shift, characterized by the disparity in the class sets considered by $P$ and $I$, causes a category gap. Category gap represents the performance drop that arises for inputs from $I$, which cannot be processed properly due to differing class sets between $P$ and $I$, which are erroneously handled. A larger category gap makes distinguishing *unknown* samples from *transferable* samples harder, thereby worsening the robustness of the model.

From the perspective of addressing the domain gap and category gap, the main goal of UniDA is to minimize both gaps simultaneously. This aims to ensure *transferable* samples properly align with the source domain for adequate processing, while handling *unknown* samples as unprocessable exceptions.

## 3 Testbed Design

The primary objective of our research is to construct a comprehensive benchmark dataset that effectively captures the viewpoint of UniDA. To accomplish our objective, we attempt to create a diverse dataset that encompasses a range of difficulty levels and characteristics, such as domains, sentiment, or temporal change. These variations are the fundamental elements that can significantly influence the overall performance.

Specifically, we initially select datasets from multiple practical domains and approximate the adaptation difficulty by quantifying different shifts with our newly proposed metrics. In the following subsections, we provide an in-depth explanation of our dataset along with the analysis of our benchmarks.

### 3.1 Quantifying Different Shifts

As the extent of both domain and category gaps significantly influences the overall adaptation complexity, it is essential to quantify these gaps when designing the dataset for evaluation. Unfortunately, existing literature has not devised a clear-cut and quantitative measure for assessing domain and category gaps. Therefore, we endeavoured to define measures that can aptly approximate the two types of gaps.

**Performance Drop Rate (PDR)** measures the degree of domain gap by assessing the absolute difficulty of the dataset itself and the performance drop caused by the shift from $P$ to $H$. Specifically, we fine-tune *bert-base-uncased* on the source train set and evaluate its test set accuracy $acc_s$ from the same distribution. Leveraging the same model trained on the source domain, we then measure the accuracy of the target test set $acc_t$. We measure the performance degradation caused by the distributional shift by measuring $acc_s - acc_t$. Since the significance of the performance drop may vary depending on the source performance, we normalize the result with the source performance and measure the proportion of the performance degradation. A more significant drop rate indicates a greater decline in performance, considering the source domain performance. Formally, PDR for a source domain $s$ and a target domain $t$ can be measured as follows:

$$\text{PDR}_{s,t} = 100 \times \frac{acc_s - acc_t}{acc_s} \qquad (1)$$

**Distinction Difficulty Score (DDS)** is measured to estimate the difficulty of distinguishing between $H$ and $I$, which, in other words, measures the difficulty of handling the category shift. We utilized the same model trained on the source domain and extracted the [CLS] representations of the source inputs. We estimated the source distribution, assuming the extracted representations follow the multivariate normal distribution. We then extracted [CLS] representations of target distribution inputs from the same model and measured the Mahalanobis distance between the source distribution. Using the distance, we measured the *Area Under the ROC Curve* (AUC) as a metric for discerning $I$ and $H$. AUC values closer to 1 indicate the ease of discerning *unknown* inputs from the *transferable* inputs. Since we focus on quantifying the difficulty in distinguishing the two, we subtract the AUC from 1 to derive our final measure of interest. For

the source domain $s$, the target domain $t$, and AUC as $\text{AUC}_{s,t}$, DDS can be measured as:

$$\text{DDS}_{s,t} = 100 \times (1 - \text{AUC}_{s,t}) \qquad (2)$$

## 3.2 Implementation of Different Shifts

To construct a representative testbed for UniDA, it is essential to illustrate domain and category shifts. To exhibit domain shift, we delineated domains from various perspectives. This involves explicit factors such as temporal or sentiment and implicit definitions based on the composition of the class set. Detailed formation of domains for each dataset is stipulated in Section 3.3.

To establish category shifts, the source and the target domain must have a set of common classes $C$ and a set of their own private classes, $\bar{C}_s$ and $\bar{C}_t$, respectively. We followed previous works (You et al., 2019; Fu et al., 2020) by sorting the class name in alphabetical order and selecting the first $|C|$ classes as common, the subsequent $|\bar{C}_s|$ as source private, and the rest as target private classes. The class splits for each dataset are stated as $|C|/|\bar{C}_s|/|\bar{C}_t|$ in the main experiments.

## 3.3 Dataset Details

We focused on text classification tasks for our experiments. Four datasets were selected from multiple widely used classification domains in NLP, such as topic classification, sentiment analysis, and intent classification. We reformulated the datasets so that our testbed could cover diverse adaptation scenarios.

**Huffpost News Topic Classification (Huffpost)** (Misra, 2022) contains Huffpost news headlines spanning from 2012 to 2022. The task is to classify news categories given the headlines. Using the temporal information additionally provided, we split the dataset year-wise from 2012 to 2017, treating each year as a distinct domain. We selected the year 2012 as the source domain, with the subsequent years assigned as the target domains, creating 5 different levels of temporal shifts.

**Multilingual Amazon Reviews Corpus (Amazon)** (Keung et al., 2020) includes product reviews that are commonly used to predict star ratings based on the review, and additional product information is provided for each review. We have revised the task to predict the product information given the reviews and utilized the star ratings to define sentiment domains. Reviews with a star rating of 1 or 2 are grouped as negative sentiment, and those with

| Dataset | Huffpost (2013) | Huffpost (2014) | Huffpost (2015) | Huffpost (2016) |
|---|---|---|---|---|
| PDR | 7.87 | 20.69 | 20.47 | 25.00 |
| DDS | 19.67 | 27.79 | 25.71 | 31.19 |

| Dataset | Huffpost (2017) | CLINC-150 | MASSIVE | Amazon |
|---|---|---|---|---|
| PDR | 33.62 | 11.59 | **36.14** | 15.45 |
| DDS | 31.74 | 13.45 | 23.92 | **36.04** |

Table 1: PDR and DDS values of each datasets. The **largest value** of PDR and DDS is highlighted in bold.

a rating of 4 or 5 are categorized as positive. We exclude 3-star reviews, considering them neutral.

**MASSIVE** (FitzGerald et al., 2022) is a hierarchical dataset for intent classification. The dataset consists of 18 first-level and 60 second-level classes. Each domain is defined as a set of classes, including private classes exclusive to a specific domain and common classes shared across domains. We divided the common first-level class into two parts based on second-level classes to simulate domain discrepancy. The half of the divided common class represents the source domain while the other half represents the target domain. We assume that the second-level classes within the same first-level class share a common feature and thus can be adapted.

**CLINC-150** (Larson et al., 2019) is widely used for intent classification in OOD detection. The dataset consists of 150 second-level classes over 10 first level-classes and a single out-of-scope class. The domain is defined in the same way as MASSIVE.

## 3.4 Dataset Analysis

In this section, we intend to validate whether our testbed successfully demonstrates diverse adaptation difficulties, aligning with our original motivation. We assess adaptation difficulty from domain and category gap perspectives, each approximated by PDR and DDS, respectively.

The results of PDR and DDS are reported in Table 1. The result shows diverse PDR values ranging from 7 to 36 points, indicating various degrees of domain gap across the datasets. MASSIVE measured the most considerable domain gap, while Huffpost (2013) demonstrated the most negligible domain gap among the datasets. Additionally, our testbed covers a wide range of category gaps, indicated by the broad spectrum of DDS values. Specifically, Amazon exhibits a significantly high

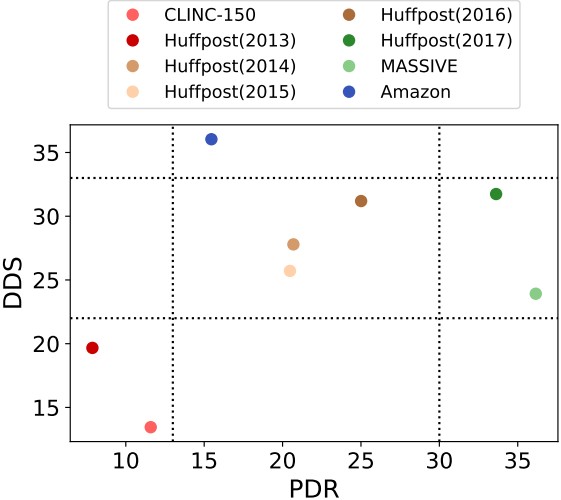

Figure 2: The visualization of adaptation difficulty for each dataset in terms of domain gap (PDR) and category gap (DDS). We categorized the dataset into 4 distinct groups based on the adaptation complexity. Best viewed in color.

DDS value, representing an extremely challenging scenario of differentiating the *unknown* samples from the *transferable* samples.

We consolidate the two indicators to measure the coverage of different adaptation complexity of our proposed testbed. We visualized the datasets considering the PDR and DDS; metrics on the respective axes. Figure 2 is the visualization of the adaptation complexity for each dataset factored by PDR and DDS. We grouped the datasets into four distinct clusters based on the plotted distribution. Datasets closer to the lower-left corner, CLINC-150 and Huffpost (2013), are easy adaptation scenarios with minor domain and category gaps. Datasets plotted in the center, Huffpost (2014, 2015, 2016), presents moderate difficulty. Amazon suffers from a category gap significantly, while Huffpost (2017) and MASSIVE demonstrate a notable domain gap, yielding high adaptation complexity. The results validate that our testbed embodies a diverse range of adaptation difficulties as intended.

## 4 Experimental Setting

### 4.1 Compared Methods

We compare several domain adaptation methods on our proposed testbed. We selected two previous state-of-the-art closed-set Domain Adaptation (CDA) methods, UDALM (Karouzos et al., 2021)

and AdSPT (Wu and Shi, 2022), under the assumption that all the inputs from the target domain are *transferable* without considering *unknown* classes. Two previous state-of-the-art UniDA methods were selected, OVANet (Saito and Saenko, 2021) and UniOT (Chang et al., 2022), which are fully optimized to handle UniDA scenarios in the vision domain. We also conducted experiments with additional baseline methods such as DANN (Ganin et al., 2016a), UAN (You et al., 2019), and CMU (Fu et al., 2020). However, the performance was subpar compared to the selected methods, exhibiting a similar tendency. Hence, we report the additional results in Appendix A. For the backbone of all the methods, we utilized *bert-base-uncased* (Devlin et al., 2019) and used the [CLS] representation as the input feature. Implementation details are stipulated in Appendix B.

### 4.2 Thresholding Method

Since CDA methods are not designed to handle *unknown* inputs, additional techniques are required to discern them. A straightforward yet intuitive approach to detecting *unknown* inputs is applying a threshold for the output of the scoring function. The scoring function reflects the appropriateness of the input based on the extracted representation. If the output of the scoring function falls below the threshold, the instance is classified as *unknown*. We sequentially apply thresholding after the adaptation process.[3] Formally, for an input $x$, categorical prediction $\hat{y}$, threshold value $w$, and a scoring function $f_{score}$, the final prediction is made as:

$$y(x) = \begin{cases} \text{argmax}(\hat{y}), & \text{if } f_{score}(x) > w \\ \text{unknown}, & \text{otherwise.} \end{cases} \quad (3)$$

We utilize Maximum Softmax Probability (MSP) as the scoring function[4] (Hendrycks and Gimpel, 2017). Following the OOD detection literature, the value at the point of 95% from the sorted score values was selected as the threshold.

---

[3]In the case of applying the thresholding first, all the inputs from the target domain would be classified as OOD. If all the target inputs were classified as OOD, the criteria for discerning *transferable* and *unknown* inputs become inherently unclear. Therefore, we have only considered scenarios where thresholding is applied after the adaptation.

[4]In addition to MSP, we have applied various scoring functions such as cosine similarity and Mahalanobis distance, but using MSP achieved the best performance. Results for other thresholding methods have been included in the Appendix A.

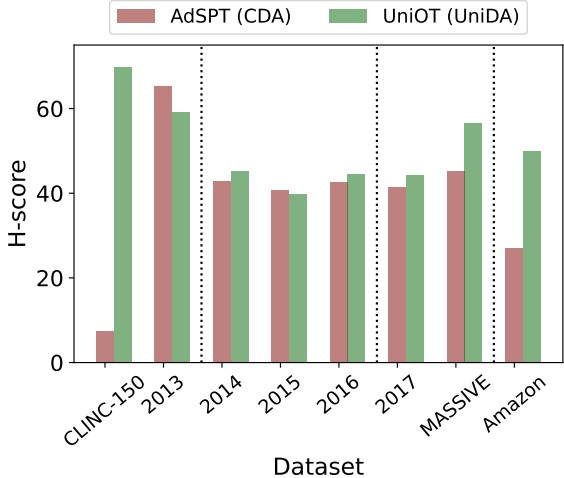

Figure 3: H-score results of AdSPT and UniOT on all the datasets. The preferred method varies depending on the adaptation complexity.

| Huffpost (2012 → 2013) (3 / 4 / 4) | | | |
|---|---|---|---|
| Method | $acc_C$ | $acc_{\bar{C}_t}$ | H-score |
| UDALM | 52.74 ±2.92 | 58.55 ±6.03 | 55.15 ±2.37 |
| AdSPT | 55.05 ±2.11 | 80.66 ±2.99 | **65.38** ±1.00 |
| OVANet | 65.11 ±0.60 | 24.91 ±6.75 | 35.64 ±6.70 |
| UniOT | 53.76 ±1.10 | 65.86 ±3.57 | 59.14 ±1.14 |
| CLINC-150 (4 / 3 / 3) | | | |
| Method | $acc_C$ | $acc_{\bar{C}_t}$ | H-score |
| UDALM | 74.92 ±6.11 | 69.91 ±13.98 | **71.28** ±4.62 |
| AdSPT | 3.96 ±2.17 | 98.69 ±0.36 | 7.55 ±3.95 |
| OVANet | 83.49 ±0.62 | 31.24 ±1.70 | 45.45 ±1.88 |
| UniOT | 64.14 ±9.14 | 77.36 ±3.94 | 69.88 ±6.11 |

Table 2: Experimental results on CLINC-150 and Huffpost (2013), which is a relatively easy adaptation scenario. The **best method** with the highest H-score is in bold, and the second-best method is underlined.

## 4.3 Evaluation Protocol

The goal of UniDA is to properly process the *transferable* inputs and detect the *unknown* inputs simultaneously, consequently making both the *transferable* and the *unknown* accuracies crucial metrics. We applied H-score (Fu et al., 2020) as the primary evaluation metric to integrate both evaluation metrics. H-score is the harmonic mean between the accuracy of common class $acc_C$ and *unknown* class $acc_{\bar{C}_t}$, where $acc_C$ is the accuracy over the common class set $C$ and $acc_{\bar{C}_t}$ is the accuracy predicting the *unknown* class. The model with a high H-score is considered robust in the UniDA setting, indicating its proficiency in both adaptation (high $acc_C$) and OOD detection (high $acc_{\bar{C}_t}$). Formally, the H-score can be defined as :

$$H_{score} = 2 \cdot \frac{acc_C \cdot acc_{\bar{C}_t}}{acc_C + acc_{\bar{C}_t}} \quad (4)$$

Although the H-score serves as an effective evaluation criterion for UniDA, we also report $acc_C$ and $acc_{\bar{C}_t}$ to provide a comprehensive assessment. We report the averaged results and standard deviations over four runs for all experiments.

## 5 Experimental Results

### 5.1 Overview

We conduct evaluations based on the clusters defined in Section 3.4 and analyze how the results vary depending on the adaptation complexity. Figure 3 presents an overview of the H-score results for the best-performing method from each CDA

and UniDA approach: AdSPT representing CDA and UniOT representing UniDA. Despite an outlier caused by unstable thresholding in CLINC-150, the overall trend demonstrates that AdSPT manifests comparable performance in less complex scenarios, while UniOT exhibits superior performance towards challenging scenarios. These trends align with the findings of other methods that are not depicted in the figure.

### 5.2 Detailed Results

Table 2 demonstrates the results of relatively easy adaptation scenarios. CDA methods demonstrate performance that is on par with, or even superior to, UniDA methods. The results appear counterintuitive, as CDA methods are designed without considering *unknown* samples. Specifically, UDALM outperforms all the UniDA methods in CLINC-150 and performs comparable or even better in Huffpost (2013). AdSPT exhibits the best performance in Huffpost (2013). However, AdSPT suffers a significant performance drop in CLINC-150, as we speculate this result is due to the inherent instability of the thresholding method. The misguided threshold classifies the majority of the inputs as *unknown*, which leads to a very high $acc_{\bar{C}_t}$, but significantly reduces the $acc_C$. This inconsistency also leads to a high variance of $acc_{\bar{C}_t}$ for all the CDA methods.

In the case of moderate shifts, no particular method decisively stands out, as presented by Table 3. In all cases, AdSPT and UniOT present

| Huffpost (2012 → 2014) (3 / 4 / 4) | | | |
|---|---|---|---|
| Method | $acc_C$ | $acc_{\bar{C}_t}$ | H-score |
| UDALM | 29.90 ±1.89 | 50.00 ±17.49 | 34.14 ±5.07 |
| AdSPT | 29.64 ±4.51 | 80.03 ±6.09 | 42.93 ±3.41 |
| OVANet | 45.85 ±2.62 | 33.96 ±6.78 | 36.02 ±3.95 |
| UniOT | 33.49 ±4.79 | 71.44 ±6.19 | **45.23** ±3.60 |

| Huffpost (2012 → 2015) (3 / 4 / 4) | | | |
|---|---|---|---|
| Method | $acc_C$ | $acc_{\bar{C}_t}$ | H-score |
| UDALM | 25.15 ±2.60 | 61.79 ±8.88 | 35.56 ±3.15 |
| AdSPT | 29.08 ±5.39 | 73.04 ±12.71 | **40.78** ±3.42 |
| OVANet | 43.91 ±2.09 | 35.70 ±4.41 | 39.21 ±2.43 |
| UniOT | 27.16 ±3.40 | 75.50 ±2.68 | 39.82 ±3.39 |

| Huffpost (2012 → 2016) (3 / 4 / 4) | | | |
|---|---|---|---|
| Method | $acc_C$ | $acc_{\bar{C}_t}$ | H-score |
| UDALM | 29.14 ±1.59 | 60.87 ±6.87 | 39.28 ±1.44 |
| AdSPT | 28.84 ±3.57 | 82.52 ±4.22 | 42.55 ±3.27 |
| OVANet | 44.30 ±2.89 | 33.49 ±4.79 | 38.03 ±3.79 |
| UniOT | 33.09 ±3.62 | 69.27 ±3.83 | **44.60** ±2.85 |

Table 3: Experimental results on Huffpost (2014, 2015, 2016), which has a moderate complexity for adaptation. The **best method** with the highest H-score is in bold, and the second-best method is underlined.

| Amazon (11 / 10 / 10) | | | |
|---|---|---|---|
| Method | $acc_C$ | $acc_{\bar{C}_t}$ | H-score |
| UDALM | 47.69 ±1.47 | 15.10 ±2.30 | 22.85 ±2.65 |
| AdSPT | 30.26 ±3.52 | 30.93 ±26.14 | 27.12 ±11.03 |
| OVANet | 44.50 ±0.84 | 42.49 ±3.95 | 43.38 ±1.83 |
| UniOT | 38.60 ±1.08 | 70.98 ±7.76 | **49.90** ±2.08 |

Table 4: Experimental results on Amazon, which has a significant influence of category gap. The **best method** with the highest H-score are in bold, and the second-best method is underlined.

| MASSIVE (8 / 5 / 5) | | | |
|---|---|---|---|
| Method | $acc_C$ | $acc_{\bar{C}_t}$ | H-score |
| UDALM | 38.33 ±4.88 | 41.29 ±15.14 | 39.28 ±9.69 |
| AdSPT | 36.78 ±2.65 | 59.87 ±13.47 | 45.32 ±6.20 |
| OVANet | 59.04 ±0.69 | 39.28 ±2.14 | 47.12 ±1.47 |
| UniOT | 44.01 ±0.20 | 79.19 ±3.52 | **56.52** ±3.49 |

| Huffpost (2012 → 2017) (3 / 4 / 4) | | | |
|---|---|---|---|
| Method | $acc_C$ | $acc_{\bar{C}_t}$ | H-score |
| UDALM | 32.85 ±1.05 | 43.08 ±6.76 | 37.08 ±2.82 |
| AdSPT | 35.74 ±3.33 | 53.04 ±16.02 | 41.36 ±5.04 |
| OVANet | 48.37 ±4.12 | 35.03 ±3.60 | 40.46 ±2.32 |
| UniOT | 31.57 ±1.92 | 75.85 ±9.63 | **44.36** ±1.27 |

Table 5: Experimental results on MASSIVE and Huffpost (2017), which demonstrates high domain gap. The **best method** with the highest H-score is in bold, and the second-best method is underlined.

the best performance with a marginal difference, making it inconclusive to determine a superior approach. Despite the relatively subpar performance, UDALM and OVANet also exhibit similar results. Still, it is notable that CDA methods, which are not inherently optimized for UniDA settings, show comparable results.

The result of Amazon, in which the category gap is most prominent, is reported in Table 4. UniDA methods exhibit substantially superior performance to CDA methods. In particular, while the difference in $acc_C$ is marginal, there exists a substantial gap of up to 55 points in $acc_{\bar{C}_t}$. As the category gap intensifies, we observe the decline in the performance of CDA methods, which are fundamentally limited by the inability to handle *unknown* inputs.

Finally, the results of Huffpost (2017) and MASSIVE, which exhibit a high domain gap, are reported in Table 5. The result indicates that UniDA methods consistently display superior performance in most cases. However, the divergence between the approaches is relatively small compared to Amazon. UniOT demonstrates the best performance in all datasets, with OVANet's slightly lower performance. AdSPT demonstrates marginally better performance than OVANet in Huffpost (2017),

but the gap is marginal. Even though CDA methods pose comparable performance, UniDA methods demonstrate better overall performance.

## 5.3 Impact of Threshold Values

The selection of the threshold value considerably influences the performance of CDA methods. In order to probe the impact of the threshold values on the performance, we carry out an analysis whereby different threshold values are applied to measure the performance of the methods.

The results are demonstrated in Figure 4. In cases of low or moderate adaptation complexity, such as CLINC-150 and Huffpost (2013, 2014, 2015, 2016), CDA methods demonstrate the potential to outperform UniDA methods when provided an appropriate threshold. However, as the adaptation complexity intensifies, such as Huffpost (2017), MASSIVE, and Amazon, UniDA methods outperform CDA methods regardless of the selected threshold. These observations align seam-

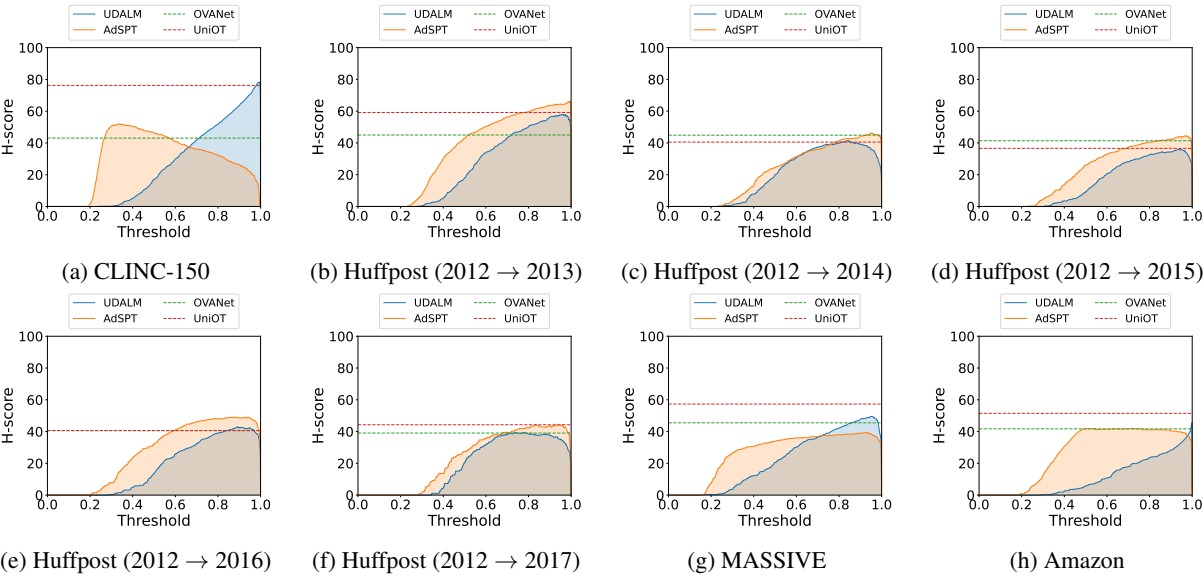

Figure 4: H-score performance with different threshold values. Results of UniDA methods are visualized as a horizontal line for comparison.

lessly with the findings from Section 4.3 that underscore the proficiency of UniDA methods in managing challenging adaptation scenarios. Additionally, it should be noted that determining the optimal threshold is particularly challenging in the absence of supervision from the target domain. Therefore, the best performance should be considered upperbound of the CDA methods.

# 6 Related Work

## 6.1 Domain Adaptation

The studies in the field of DA in NLP primarily assumes a closed-set environment, which the source and the target domain share the same label space. CDA research predominantly concentrated on learning domain invariant features (Blitzer et al., 2006; Pan et al., 2010; Ben-David et al., 2020; Ganin and Lempitsky, 2015; Du et al., 2020) for effective adaptation. With the advent of pretrained language models (PLMs), CDA methods have evolved to effectively leverage the capabilities of PLMs. Techniques such as masked language modeling (Karouzos et al., 2021) or soft-prompt with adversarial training (Wu and Shi, 2022) have shown promising results. However, the closed-set assumption has a fundamental drawback as it may leave the models vulnerable when exposed to data from an unknown class.

To mitigate such issue, a new line of work named UniDA (You et al., 2019) was proposed which assumes no prior knowledge about the target domain.

You et al. (2019) quantifies sample-level transferability by using of uncertainty and domain similarity. Following the work, Fu et al. (2020) calibrates multiple uncertainty measures to handle such an issue. Saito and Saenko (2021) apply a one-vs-all classifier to minimize inter-class distance and classify unknown classes. More recently, Chang et al. (2022) applied Optimal Transport and further expanded the task to discovering private classes. Other recent works focus on utilizing mutually nearest neighbor samples (Chen et al., 2022a,c) or leveraging source prototypes with target samples (Chen et al., 2022b; Kundu et al., 2022). Despite the practicality of UniDA, its application in the NLP domain has barely explored.

## 6.2 Out-of-Distribution Detection

The early exploration of OOD detection focused on training supervised detectors (Dhamija et al., 2018; Lee et al., 2018a; Jiang et al., 2018). However, since obtaining labeled OOD samples is impractical, recent OOD detection research has shifted towards unsupervised methods, such as generating pseudo-OOD data (Chen and Yu, 2021; Zheng et al., 2020), utilizing self-supervised learning (Moon et al., 2021; Manolache et al., 2021; Li et al., 2021; Zeng et al., 2021; Zhan et al., 2021; Cho et al., 2022), and measuring uncertainty through scoring functions for input instances (Hendrycks and Gimpel, 2017; Lee et al., 2018b; Liu et al., 2020; Tack et al., 2020). While these methods have shown effectiveness, OOD detection is limited in

that it does not offer opportunities for adaptation.

# 7 Conclusion and Future Work

In this study, we present a testbed for evaluating UniDA in the field of NLP. The testbed is designed to exhibit various levels of domain and category gaps through different datasets. Two novel metrics, PDR and DDS, were proposed which can measure the degree of domain and category gap, respectively. We assessed UniDA methods and the heuristic combination of CDA and OOD detection in our proposed testbed. Experimental results show that UniDA methods, initially designed for the vision domain, can be effectively transferred to NLP. Additionally, CDA methods, which are not fully optimized in UniDA scenario, produce comparable results in certain circumstances.

Recent trends in NLP focus on Large Language Models (LLMs) of their significant generalization abilities. However, the robustness of LLMs from the perspective of UniDA remains uncertain. As part of our future work, we assess the performance and the capabilities of LLMs from a UniDA viewpoint.

# Limitations

**Limited coverage of the evaluated model sizes**
The evaluation was conducted only with models of limited size. Moreover, there is a lack of zero-shot and few-shot evaluations for large language models (LLMs) that have recently emerged with remarkable generalization capabilities. The evaluation of LLMs is currently being considered as a top priority for our future work, and based on preliminary experiments, the results were somewhat unsatisfactory compared to small models with basic tuning for classification performance. In this regard, recent research that evaluated LLMs for classification problems (such as GLUE) also reported that the performance is not yet comparable to task-specifically tuned models. Considering the limitations of LLMs in terms of their massive resource usage and the fact that tuning small models still outperforms them in a task-specific manner, the findings from this study are still considered highly valuable in the NLP community.

**Limited scope of the tasks** Our proposed testbed is restricted to text classification tasks only. The majority of existing research on DA and OOD also focuses on classification. This selective task preference is primarily due to the challenge of defining concepts such as domain shifts and category shifts in generative tasks. However, in light of recent advancements in the generative capabilities of models, handling distributional shifts in generative tasks is indubitably an essential problem that needs to be addressed in our future work.

# Acknowledgement

This work was mainly supported by SNU-NAVER Hyperscale AI Center and partly supported by Institute of Information & communications Technology Planning & Evaluation (IITP) grant funded by the Korea government (MSIT) [No.2020-0-01373, Artificial Intelligence Graduate School Program (Hanyang University), NO.2021-0-02068, Artificial Intelligence Innovation Hub (Artificial Intelligence Institute, Seoul National University)] and Korea Evaluation Institute of Industrial Technology (KEIT) grant funded by the Korea government (MOTIE).

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

# A  Full Experimental Results

## A.1  UniDA Results

Table 7 is the full results of UniDA methods in our proposed testbed. Baseline methods such as UAN (You et al., 2019) and CMU (Fu et al., 2020) are included in the results. We can observe that UniDA methods do not always retrain the same level of applicability in NLP. Specifically, UAN and CMU utilize a fixed threshold defined in the vision domain. While CMU remains fully compatible in the NLP domain, UAN struggles to apply effectively, as it fails to detect *unknown* samples.

## A.2  CDA Results

In this section, we demonstrate the experimental results of CDA methods with two additional scoring functions: cosine similarity and Mahalanobis distance. The threshold value was selected based on the score from the scoring functions, using the same approach as the main experiment. Also, we report the results of DANN (Ganin et al., 2016a) and source-only fine-tuning which was left out from the main experiment. In some cases, source-only fine-tuning outperforms other adaptation methods, which is also observed in the vision domain (You et al., 2019).

1. **Cosine Similarity** (Tack et al., 2020) calculates the cosine similarity score between the test input and the train input. The score is selected as the cosine similarity between the input and the nearest neighbor. The results are reported in Table 8.

2. **Mahalanobis Distance** (Lee et al., 2018b) is the distance of the test sample to each class distribution. The representation is assumed to follow the multivariate normal distributions. The distance between the nearest class distribution is used as the score. The results are demonstrated in Table 9

Additionally, the full experimental results of MSP thresholding are presented in Table 10.

| CDA Methods | BERT | DANN | UDALM | AdSPT |
|---|---|---|---|---|
| Huffpost (2013) | 5e-5 | 5e-5 | 1e-4 | 5e-5 |
| Huffpost (2014) | 5e-5 | 5e-5 | 1e-4 | 5e-5 |
| Huffpost (2015) | 5e-5 | 5e-5 | 1e-4 | 5e-5 |
| Huffpost (2016) | 5e-5 | 5e-5 | 1e-4 | 5e-5 |
| Huffpost (2017) | 5e-5 | 1e-4 | 1e-4 | 5e-5 |
| CLINC-150 | 1e-4 | 5e-4 | 5e-4 | 1e-5 |
| MASSIVE | 5e-5 | 5e-5 | 5e-5 | 5e-5 |
| Amazon | 1e-5 | 5e-6 | 5e-5 | 1e-5 |

| UniDA Methods | UAN | CMU | OVANet | UniOT |
|---|---|---|---|---|
| Huffpost (2013) | 5e-5 | 5e-5 | 5e-5 | 1e-4 |
| Huffpost (2014) | 5e-5 | 5e-5 | 5e-5 | 1e-4 |
| Huffpost (2015) | 1e-4 | 1e-4 | 5e-5 | 5e-5 |
| Huffpost (2016) | 1e-4 | 5e-5 | 5e-5 | 1e-4 |
| Huffpost (2017) | 1e-4 | 5e-5 | 5e-5 | 1e-4 |
| CLINC-150 | 5e-6 | 1e-5 | 5e-6 | 5e-5 |
| MASSIVE | 5e-5 | 5e-5 | 5e-5 | 1e-4 |
| Amazon | 5e-5 | 5e-5 | 5e-5 | 5e-5 |

Table 6: Learning rates for each methods. The learning rate was selected from 5e-4, 1e-4, 5e-5, 1e-5, and 1e-6 with the best evaluation performance from the source domain.

## B  Implementation Details

For the experiments, we adopt a 12-layer pre-trained language model *bert-base-uncased* (Devlin et al., 2019) as the backbone of all the methods. We utilized the [CLS] representation as the input feature. AdamW optimizer (Loshchilov and Hutter, 2019) was used for all the experiments with a batch size of 32. We selected the best learning rate among 5e-4, 1e-4, 5e-5, 1e-5, and 5e-6. The learning rate for each method is reported in Table 6. The model was trained for 10 epochs with an early stopping on the accuracy of the source domain's evaluation set. All the experiments were implemented with Pytorch (Paszke et al., 2019) and Huggingface Transformers library (Wolf et al., 2020). The experiments take an hour on a single Tesla V100 GPU.

## C  Ablation on Different Class Splits

For the main experiment, we utilized class names as the criterion to implement the category gap. However, this may only show the specific scenario of the category gap. To provide a more comprehensive analysis, we also report the results when the class set is randomly split. We utilized CLINC-150 and MASSIVE dataset for the ablation study, and MSP thresholding was applied for CDA methods. We conducted three experiments, each with a different class split, and for every split, we reported

the average results of three different runs.

Table 11 is the results of the experiments. Due to the changes in the class set to be predicted, the task difficulty varies, resulting in differences in absolute performances. However, when comparing the relative performance between different methods, we can observe that they exhibit consistent trends regardless of the class split.

## D  Receiver Operating Characteristic (ROC) Curve

To measure Distinction Difficulty Score (DDS), we calculated the AUROC and subtracted from 1. Figure 5 is the ROC curve of discerning *unknown* inputs from the *transferable* inputs for our proposed datasets. The closer the ROC curve is to the upper-left corner, it indicates that it is easier to distinguish between *unknown* and *transferable* inputs.

| Dataset | Huffpost (2012 → 2013) (3 / 4 / 4) | | | Huffpost (2012 → 2014) (3 / 4 / 4) | | |
|---|---|---|---|---|---|---|
| Method | $acc_C$ | $acc_{\bar{C}_t}$ | H-score | $acc_C$ | $acc_{\bar{C}_t}$ | H-score |
| UAN | 73.58 ±5.76 | 0.00 ±0.00 | 0.00 ±0.00 | 36.38 ±3.04 | 0.00 ±0.00 | 0.00 ±0.00 |
| CMU | 58.18 ±1.88 | 26.89 ±4.29 | 36.76 ±4.22 | 33.13 ±3.35 | 39.98 ±3.71 | 36.02 ±1.46 |
| OVANet | 65.11 ±0.60 | 24.91 ±6.75 | 35.64 ±6.70 | 45.85 ±2.62 | 33.96 ±6.78 | 38.56 ±3.95 |
| UniOT | 53.76 ±1.10 | 65.86 ±3.57 | **59.14** ±1.14 | 33.49 ±4.79 | 71.44 ±6.19 | **45.23** ±3.60 |

| Dataset | Huffpost (2012 → 2015) (3 / 4 / 4) | | | Huffpost (2012 → 2016) (3 / 4 / 4) | | |
|---|---|---|---|---|---|---|
| Method | $acc_C$ | $acc_{\bar{C}_t}$ | H-score | $acc_C$ | $acc_{\bar{C}_t}$ | H-score |
| UAN | 34.17 ±6.00 | 0.00 ±0.00 | 0.00 ±0.00 | 42.53 ±16.52 | 0.00 ±0.00 | 0.00 ±0.00 |
| CMU | 39.57 ±3.54 | 17.98 ±6.30 | 24.02 ±5.57 | 41.80 ±2.08 | 30.09 ±3.51 | 34.84 ±1.97 |
| OVANet | 43.91 ±2.09 | 35.70 ±4.41 | 39.21 ±2.43 | 44.30 ±2.89 | 33.49 ±4.79 | 38.03 ±3.79 |
| UniOT | 27.16 ±3.40 | 75.50 ±2.68 | **39.82** ±3.39 | 33.09 ±3.62 | 69.27 ±3.83 | **44.60** ±2.85 |

| Dataset | Huffpost (2012 → 2017) (3 / 4 / 4) | | | CLINC-150 (4 / 3 / 3) | | |
|---|---|---|---|---|---|---|
| Method | $acc_C$ | $acc_{\bar{C}_t}$ | H-score | $acc_C$ | $acc_{\bar{C}_t}$ | H-score |
| UAN | 22.74 ±10.63 | 0.00 ±0.00 | 0.00 ±0.00 | 48.99 ±14.49 | 0.00 ±0.00 | 0.00 ±0.00 |
| CMU | 34.87 ±3.14 | 55.79 ±2.71 | 42.86 ±2.72 | 60.70 ±1.02 | 36.89 ±1.52 | 45.87 ±1.09 |
| OVANet | 48.37 ±4.12 | 35.03 ±3.60 | 40.46 ±2.32 | 83.49 ±0.62 | 31.24 ±1.70 | 45.45 ±1.88 |
| UniOT | 31.57 ±1.93 | 75.85 ±9.63 | **44.36** ±1.27 | 64.14 ±9.14 | 77.36 ±3.94 | **69.88** ±6.11 |

| Dataset | MASSIVE (8 / 5 / 5) | | | Amazon (11 / 10 / 10) | | |
|---|---|---|---|---|---|---|
| Method | $acc_C$ | $acc_{\bar{C}_t}$ | H-score | $acc_C$ | $acc_{\bar{C}_t}$ | H-score |
| UAN | 13.00 ±7.18 | 0.00 ±0.00 | 0.00 ±0.00 | 35.40 ±3.44 | 0.00 ±0.00 | 0.00 ±0.00 |
| CMU | 37.38 ±1.59 | 10.33 ±3.06 | 16.03 ±3.84 | 44.30 ±0.90 | 39.95 ±9.67 | 41.48 ±5.10 |
| OVANet | 59.04 ±0.69 | 39.25 ±2.14 | 47.12 ±1.47 | 44.50 ±0.84 | 42.49 ±3.95 | 43.38 ±1.83 |
| UniOT | 44.01 ±3.52 | 79.19 ±4.14 | **56.52** ±3.49 | 38.60 ±1.08 | 70.98 ±7.76 | **49.90** ±2.08 |

Table 7: Experimental results of UniDA methods in the proposed testbed. For each dataset, the **best method** with the highest H-score is in bold and the second-best method is underlined.

| Dataset | Huffpost (2012 → 2013) (3 / 4 / 4) | | | Huffpost (2012 → 2014) (3 / 4 / 4) | | |
|---|---|---|---|---|---|---|
| Method | $acc_C$ | $acc_{\bar{C}_t}$ | H-score | $acc_C$ | $acc_{\bar{C}_t}$ | H-score |
| BERT | 64.06 ±2.13 | 28.02 ±3.03 | **38.86** ±2.63 | 40.63 ±3.55 | 37.41 ±4.81 | **38.65** ±1.34 |
| DANN | 58.92 ±1.62 | 1.69 ±0.57 | 3.28 ±1.07 | 33.08 ±2.94 | 2.01 ±0.82 | 3.76 ±1.48 |
| UDALM | 60.12 ±2.56 | 23.94 ±3.80 | 34.12 ±4.00 | 37.97 ±1.69 | 29.27 ±6.15 | 32.84 ±4.22 |
| AdSPT | 62.71 ±0.88 | 10.89 ±3.04 | 18.41 ±4.38 | 39.57 ±3.81 | 23.88 ±3.44 | 29.74 ±3.59 |

| Dataset | Huffpost (2012 → 2015) (3 / 4 / 4) | | | Huffpost (2012 → 2016) (3 / 4 / 4) | | |
|---|---|---|---|---|---|---|
| Method | $acc_C$ | $acc_{\bar{C}_t}$ | H-score | $acc_C$ | $acc_{\bar{C}_t}$ | H-score |
| BERT | 40.75 ±2.12 | 30.90 ±1.55 | **35.13** ±1.53 | 47.07 ±2.46 | 26.15 ±3.73 | **33.42** ±2.43 |
| DANN | 31.71 ±2.49 | 3.39 ±0.92 | 6.11 ±1.57 | 36.68 ±3.86 | 2.16 ±0.80 | 4.05 ±1.44 |
| UDALM | 37.06 ±1.74 | 24.79 ±4.83 | 29.59 ±3.89 | 44.10 ±1.52 | 23.26 ±2.48 | 30.38 ±2.02 |
| AdSPT | 39.92 ±4.75 | 15.01 ±3.85 | 21.74 ±4.84 | 41.44 ±3.63 | 12.66 ±3.31 | 19.14 ±3.97 |

| Dataset | Huffpost (2012 → 2017) (3 / 4 / 4) | | | CLINC-150 (4 / 3 / 3) | | |
|---|---|---|---|---|---|---|
| Method | $acc_C$ | $acc_{\bar{C}_t}$ | H-score | $acc_C$ | $acc_{\bar{C}_t}$ | H-score |
| BERT | 48.84 ±3.16 | 29.52 ±4.67 | **36.62** ±3.91 | 74.53 ±3.02 | 71.37 ±6.54 | 72.69 ±2.48 |
| DANN | 31.30 ±2.34 | 9.32 ±1.63 | 14.31 ±2.02 | 60.21 ±9.33 | 49.65 ±14.72 | 52.74 ±10.15 |
| UDALM | 41.08 ±3.92 | 29.10 ±7.01 | 33.48 ±3.83 | 75.99 ±3.63 | 78.11 ±5.41 | **76.90** ±2.35 |
| AdSPT | 43.02 ±0.74 | 31.07 ±4.77 | 35.93 ±3.37 | 42.94 ±5.59 | 6.96 ±3.44 | 11.57 ±4.59 |

| Dataset | MASSIVE (8 / 5 / 5) | | | Amazon (11 / 10 / 10) | | |
|---|---|---|---|---|---|---|
| Method | $acc_C$ | $acc_{\bar{C}_t}$ | H-score | $acc_C$ | $acc_{\bar{C}_t}$ | H-score |
| BERT | 51.21 ±3.62 | 62.96 ±2.06 | 56.44 ±2.69 | 44.88 ±2.10 | 14.49 ±1.25 | 21.90 ±1.68 |
| DANN | 41.55 ±9.00 | 13.74 ±3.52 | 20.62 ±5.04 | 44.58 ±0.97 | 19.61 ±1.08 | **27.21** ±0.93 |
| UDALM | 65.22 ±2.67 | 64.25 ±2.17 | **64.67** ±0.76 | 46.65 ±0.47 | 13.58 ±2.23 | 20.96 ±2.79 |
| AdSPT | 36.78 ±2.65 | 59.87 ±13.47 | 45.32 ±6.20 | 30.26 ±1.45 | 6.45 ±4.55 | 10.15 ±5.65 |

Table 8: Experimental results of CDA methods with *cosine similarity* as the scoring function. For each dataset, the **best method** with the highest H-score is in bold and the second-best method is underlined.

| Dataset | Huffpost (2012 → 2013) (3 / 4 / 4) | | | Huffpost (2012 → 2014) (3 / 4 / 4) | | |
|---|---|---|---|---|---|---|
| Method | $acc_C$ | $acc_{\bar{C}_t}$ | H-score | $acc_C$ | $acc_{\bar{C}_t}$ | H-score |
| BERT | 10.40 ±12.11 | 35.69 ±5.72 | **13.78** ±12.38 | 11.28 ±8.86 | 42.44 ±2.47 | 16.41 ±9.26 |
| DANN | 18.55 ±9.65 | 1.50 ±0.30 | 2.69 ±0.62 | 16.78 ±7.67 | 1.48 ±0.67 | 2.70 ±1.22 |
| UDALM | 7.39 ±10.12 | 27.73 ±4.25 | 9.26 ±10.18 | 4.92 ±4.14 | 34.04 ±5.66 | 8.15 ±6.52 |
| AdSPT | 18.07 ±10.66 | 7.70 ±0.36 | 10.21 ±2.49 | 17.29 ±8.06 | 18.80 ±3.47 | **16.87** ±5.39 |

| Dataset | Huffpost (2012 → 2015) (3 / 4 / 4) | | | Huffpost (2012 → 2016) (3 / 4 / 4) | | |
|---|---|---|---|---|---|---|
| Method | $acc_C$ | $acc_{\bar{C}_t}$ | H-score | $acc_C$ | $acc_{\bar{C}_t}$ | H-score |
| BERT | 11.28 ±11.10 | 37.71 ±4.18 | 15.38 ±11.27 | 10.34 ±9.85 | 31.43 ±5.76 | **14.52** ±10.93 |
| DANN | 20.09 ±5.97 | 2.93 ±1.23 | 4.88 ±1.59 | 20.79 ±10.71 | 1.92 ±0.32 | 3.44 ±0.41 |
| UDALM | 4.19 ±2.98 | 31.19 ±6.15 | 7.06 ±4.84 | 6.23 ±5.09 | 31.52 ±5.19 | 9.49 ±6.74 |
| AdSPT | 19.20 ±7.31 | 14.55 ±3.02 | **16.30** ±4.47 | 17.78 ±9.51 | 9.73 ±4.76 | 12.10 ±6.48 |

| Dataset | Huffpost (2012 → 2017) (3 / 4 / 4) | | | CLINC-150 (4 / 3 / 3) | | |
|---|---|---|---|---|---|---|
| Method | $acc_C$ | $acc_{\bar{C}_t}$ | H-score | $acc_C$ | $acc_{\bar{C}_t}$ | H-score |
| BERT | 10.31 ±13.20 | 34.04 ±5.04 | 13.10 ±13.18 | 16.67 ±21.48 | 72.56 ±5.44 | **21.87** ±25.69 |
| DANN | 16.79 ±0.94 | 7.48 ±1.62 | 10.29 ±1.63 | 1.87 ±1.36 | 41.89 ±20.06 | 3.25 ±1.89 |
| UDALM | 7.65 ±5.29 | 30.08 ±8.33 | 10.93 ±6.62 | 5.18 ±8.80 | 80.15 ±3.95 | 8.73 ±14.40 |
| AdSPT | 19.77 ±8.68 | 30.93 ±3.52 | **23.30** ±7.35 | 6.59 ±7.97 | 29.85 ±29.28 | 7.41±5.75 |

| Dataset | MASSIVE (8 / 5 / 5) | | | Amazon (11 / 10 / 10) | | |
|---|---|---|---|---|---|---|
| Method | $acc_C$ | $acc_{\bar{C}_t}$ | H-score | $acc_C$ | $acc_{\bar{C}_t}$ | H-score |
| BERT | 5.62 ±9.41 | 68.53 ±4.54 | 9.07 ±14.57 | 4.66 ±2.59 | 15.97 ±1.59 | 6.85 ±3.45 |
| DANN | 4.59 ±3.59 | 9.73 ±3.60 | 5.26 ±2.70 | 8.62 ±7.31 | 17.74 ±3.69 | **9.75** ±3.92 |
| UDALM | 2.96 ±4.90 | 62.80 ±5.02 | 5.17 ±8.36 | 2.68 ±2.80 | 19.01 ±5.49 | 4.48 ±4.22 |
| AdSPT | 5.70 ±6.41 | 54.61 ±12.44 | **9.60** ±10.75 | 4.63 ±4.66 | 9.38 ±6.68 | 5.61 ±5.65 |

Table 9: Experimental results of CDA methods with *Mahalanobis distance* as the scoring function. For each dataset, the **best method** with the highest H-score is in bold and the second-best method is underlined.

| Dataset | Huffpost (2012 → 2013) (3 / 4 / 4) | | | Huffpost (2012 → 2014) (3 / 4 / 4) | | |
|---|---|---|---|---|---|---|
| Method | $acc_C$ | $acc_{\bar{C}_t}$ | H-score | $acc_C$ | $acc_{\bar{C}_t}$ | H-score |
| BERT | 51.47 ±3.72 | 72.07 ±10.30 | 59.64 ±3.28 | 26.71 ±1.99 | 78.98 ±8.23 | 39.75 ±1.51 |
| DANN | 49.35 ±6.58 | 77.79 ±25.05 | 57.83 ±6.49 | 14.94 ±3.26 | 95.24 ±0.70 | 25.71 ±4.48 |
| UDALM | 52.74 ±2.92 | 58.25 ±6.03 | 55.15 ±2.37 | 29.90 ±1.89 | 50.00 ±17.49 | 36.14 ±5.07 |
| AdSPT | 55.05 ±2.11 | 80.66 ±2.99 | **65.38** ±1.00 | 29.64 ±4.51 | 80.03 ±6.09 | **42.93** ±3.71 |

| Dataset | Huffpost (2012 → 2015) (3 / 4 / 4) | | | Huffpost (2012 → 2016) (3 / 4 / 4) | | |
|---|---|---|---|---|---|---|
| Method | $acc_C$ | $acc_{\bar{C}_t}$ | H-score | $acc_C$ | $acc_{\bar{C}_t}$ | H-score |
| BERT | 27.35 ±2.64 | 77.13 ±2.20 | 40.30 ±2.60 | 33.12 ±1.85 | 63.65 ±3.63 | **43.53** ±1.96 |
| DANN | 16.48 ±7.90 | 89.93 ±11.51 | 26.81 ±9.33 | 11.23 ±4.13 | 95.41 ±1.23 | 19.89 ±6.52 |
| UDALM | 25.15 ±2.60 | 61.79 ±8.88 | 35.56 ±3.15 | 29.14 ±1.59 | 60.87 ±6.87 | 39.28 ±1.44 |
| AdSPT | 29.08 ±5.39 | 73.04 ±12.71 | **40.78** ±3.42 | 28.84 ±3.57 | 82.52 ±4.22 | 42.55 ±3.27 |

| Dataset | Huffpost (2012 → 2017) (3 / 4 / 4) | | | CLINC-150 (4 / 3 / 3) | | |
|---|---|---|---|---|---|---|
| Method | $acc_C$ | $acc_{\bar{C}_t}$ | H-score | $acc_C$ | $acc_{\bar{C}_t}$ | H-score |
| BERT | 33.07 ±3.90 | 65.82 ±7.47 | **43.64** ±1.71 | 64.69 ±3.68 | 72.08 ±9.82 | 67.82 ±4.15 |
| DANN | 22.84 ±2.60 | 82.46 ±4.39 | 35.66 ±2.85 | 48.02 ±4.93 | 67.48 ±19.86 | 54.90 ±9.17 |
| UDALM | 32.85 ±1.05 | 43.08 ±6.76 | 37.08 ±2.82 | 74.92 ±6.11 | 69.91 ±13.98 | **71.28** ±4.62 |
| AdSPT | 35.74 ±3.33 | 53.04 ±16.02 | 41.36 ±5.04 | 3.96 ±2.17 | 98.69 ±0.36 | 7.55 ±3.95 |

| Dataset | MASSIVE (8 / 5 / 5) | | | Amazon (11 / 10 / 10) | | |
|---|---|---|---|---|---|---|
| Method | $acc_C$ | $acc_{\bar{C}_t}$ | H-score | $acc_C$ | $acc_{\bar{C}_t}$ | H-score |
| BERT | 30.04 ±2.74 | 68.95 ±5.53 | 41.79 ±3.20 | 44.98 ±2.32 | 12.66 ±1.47 | 19.70 ±1.69 |
| DANN | 20.37 ±11.32 | 88.69 ±5.27 | 31.77 ±13.32 | 45.85 ±0.49 | 12.52 ±0.71 | 19.66 ±0.87 |
| UDALM | 38.33 ±4.88 | 41.29 ±15.14 | 39.28 ±9.69 | 47.69 ±1.47 | 15.10 ±2.30 | 22.85 ±2.65 |
| AdSPT | 36.78 ±2.65 | 59.87 ±13.47 | **45.32** ±6.20 | 30.26 ±3.52 | 30.93 ±26.14 | **27.12** ±11.03 |

Table 10: Experimental results of CDA methods with *Maximum Softmax Probability* as the scoring function. For each dataset, the **best method** with the highest H-score is in bold and the second-best method is underlined.

| Dataset | CLINC-150 (4 / 3 / 3) | | | MASSIVE (8 / 5 / 5) | | |
|---|---|---|---|---|---|---|
| Runs | 1 | 2 | 3 | 1 | 2 | 3 |
| BERT | 44.59 ±6.01 | 48.08 ±2.90 | 52.25 ±3.54 | 51.68 ±1.62 | 46.68 ±1.90 | 51.56 ±0.81 |
| DANN | 47.91 ±9.96 | 53.23 ±7.92 | 60.42 ±7.00 | 47.16 ±3.79 | 31.39 ±3.93 | 45.21 ±8.93 |
| UDALM | 49.17 ±8.75 | 55.04 ±5.31 | 52.42 ±8.20 | 55.52 ±3.00 | 42.34 ±5.88 | 51.15 ±3.62 |
| AdSPT | 31.81 ±8.63 | 48.62 ±1.85 | 46.20 ±5.22 | 45.41 ±9.04 | 38.61 ±3.90 | 46.95 ±5.22 |
| UAN | 0.00 ±0.00 | 0.00 ±0.00 | 0.00 ±0.00 | 0.00 ±0.00 | 0.00 ±0.00 | 0.00 ±0.00 |
| CMU | 40.32 ±2.32 | 38.79 ±2.19 | 40.92 ±1.77 | 21.46 ±1.62 | 18.84 ±5.26 | 28.33 ±4.29 |
| OVANet | 41.77 ±1.90 | 35.89 ±3.86 | 29.15 ±2.17 | **63.91** ±3.41 | 54.31 ±0.63 | 39.85 ±3.74 |
| UniOT | **50.94** ±7.20 | **61.66** ±1.70 | 56.51 ±3.53 | 62.07 ±2.89 | **55.11** ±2.31 | **63.59** ±3.15 |

Table 11: H-score results of UniDA methods in CLINC-150 and MASSIVE with different class splits. The **best method** with the highest H-score is in bold and the second-best method is underlined.

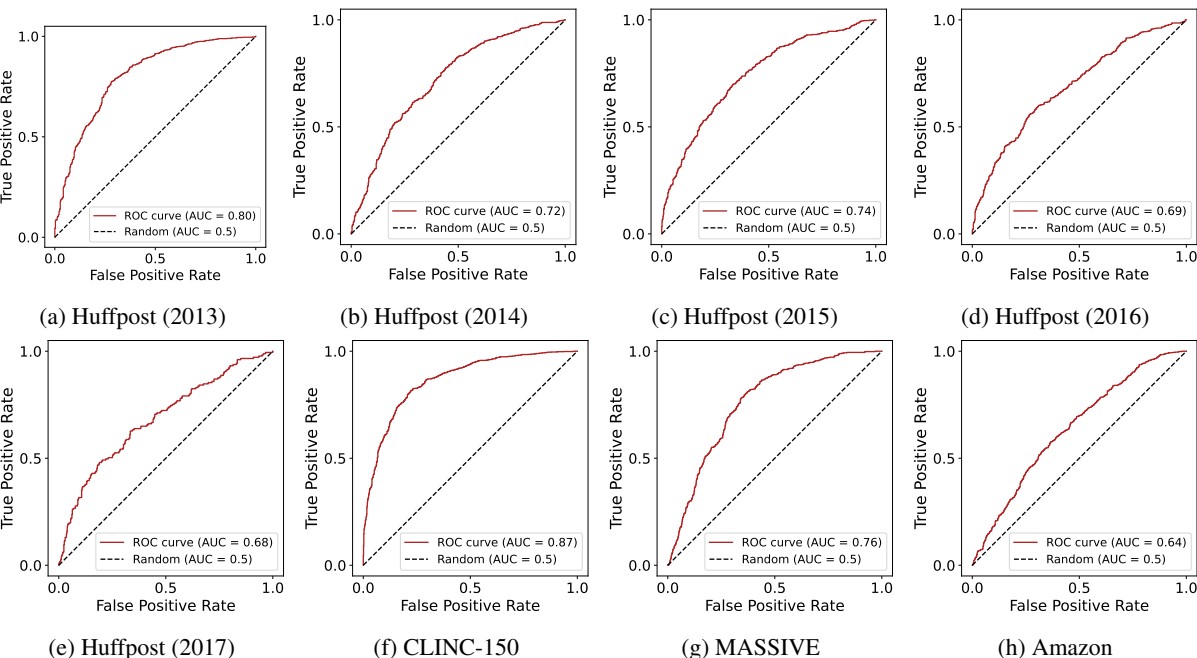

Figure 5: ROC curve of discerning *unknown* samples from the *transferable* samples. The closer the ROC curve is to the upper-left corner, it becomes easier to distinguish between the them.