# OpenReview forum: "Universal Domain Adaptation for Robust Handling of Distributional Shifts in NLP"
_EMNLP/2023/Conference — EMNLP 2023 Findings_

### Official Review · Reviewer_pfYB · 2023-07-21

**Typos Grammar Style And Presentation Improvements:** 1) Line 215
**Soundness:** 3

**Excitement:**

3: Ambivalent: It has merits (e.g., it reports state-of-the-art results, the idea is nice), but there are key weaknesses (e.g., it describes incremental work), and it can significantly benefit from another round of revision. However, I won't object to accepting it if my co-reviewers champion it.

**Paper Topic And Main Contributions:**

This paper proposes a benchmark for Universal Domain Adaptation (UniDA). When transferring from a source domain to a target domain, UniDA requires a model to deal with transferable samples properly while detecting out-of-domain samples. This problem faces two challenges: domain gap requires models to transfer across domains while category shift requires models to distinguish unknown samples from transferable samples. To evaluate the proposed benchmark, the authors propose two measurements to quantize two major challenges of UniDA. Then, extensive experiments show the performance of existing methods on the new benchmark.

**Reasons To Accept:**

1) This paper provides a new benchmark for a new challenging task. It provides a couple of datasets and a standard metric for future works.
2) The benchmark is fully investigated and can cover most scenarios of UniDA.

**Reasons To Reject:**

1) The proposed metric for the task (section 4.3) is incomprehensive. The authors claim that UniDA is challenging because it has two gaps, but the proposed H-score cannot reflect how well a model can solve one certain gap. As a result, it is also uncertain whether the claimed two challenges are all difficult for existing works or worth studying both.
2) Baseline methods are limited. The problem is a combination of DA and OOD detection so an intuitive thought is to do the OOD detection first and process the in-domain samples with a DA model. The baselines in this paper are either single DA models for NLP or UniDA models for computer vision but the combination of OOD detection model for NLP and DA model for NLP is missing.

**Edit after rebuttal**

All my concerns are being answered. But it seems like the work always follows what CV has done but lacks NLP-specific discussion.

**Reproducibility:**

4: Could mostly reproduce the results, but there may be some variation because of sample variance or minor variations in their interpretation of the protocol or method.

**Reviewer Confidence:**

4: Quite sure. I tried to check the important points carefully. It's unlikely, though conceivable, that I missed something that should affect my ratings.

---

> ### Author Rebuttal · Authors · 2023-08-28
>
> We genuinely appreciate your thoughtful feedback. Below is our response to your comments.
>
>
> **W1. Incomprehensive Metric (H-score)**
>
> We agree that the H-score has its limitations.
> However, the H-score has consistently been employed across the Universal Domain Adaptation (UniDA) literature ([1], [2], [3], [4]) and is currently perceived as the best metric for measuring the essence of the UniDA objective,
> which is to minimize both the domain and category gap simultaneously.
>
> As the reviewer clearly pointed out, examining the optimal performance of each method for *one certain gap* can also offer intriguing observations.
> We believe it could be a factor well worth considering in future UniDA research.
> Yet, the UniDA objective is not solely limited to handling a single gap - instead, it seeks to robustly handle the concurrent occurrence of both the domain and category gap.
> From this standpoint, the H-score, harmonic mean of *transferable* and *unknown* accuracy, is deemed a valid evaluation metric for UniDA.
>
>
>
> **W2. Limited Baseline Methods**
>
> We have also considered the scenario the reviewer mentioned during the research process.
> The rationale behind excluding this scenario from our baselines is the following:
>
> - The samples from the target domain are inherently *“out-of-domain”* (OOD) from the perspective of the source domain.
> - OOD detection methods, which aim to detect all out-of-domain inputs, would most likely classify all the samples from the target domain as OOD.
> - If all samples in the target domain were classified as OOD, the criteria for discerning *transferable* and *unknown* inputs become inherently unclear.
> - Hence, we decided to solely employ baselines that first apply adaptation and distinguish *unknown* samples afterward.
>
> Our work follows the early research in the UniDA literature from the vision domain ([1], [2], [5], [6]), which predominantly considers baselines that prioritize applying domain adaptation (DA).
>
>
> [1] Fu et al., Learning to Detect Open Classes for Universal Domain Adaptation. ECCV 2020
>
> [2] Li et al., Domain consensus clustering for universal domain adaptation. CVPR 2021
>
> [3] Chang et al., Unified Optimal Transport Framework for Universal Domain Adaptation. NeurIPS 2022
>
> [4] Chen et al., Evidential Neighborhood Contrastive Learning for Universal Domain Adaptation. AAAI 2022
>
> [5] You et al., Universal Domain Adaptation. CVPR 2019
>
> [6] Saito et al., Universal Domain Adaptation through Self Supervision. NeurIPS 2020

---

### Official Review · Reviewer_17io · 2023-07-29

**Soundness:** 4

**Excitement:**

4: Strong: This paper deepens the understanding of some phenomenon or lowers the barriers to an existing research direction.

**Paper Topic And Main Contributions:**

**Paper Topic:**

This paper centers around the topic of Universal Domain Adaptation (UniDA) in the field of Natural Language Processing (NLP). UniDA, which has shown promising results in the computer vision field, aims to enable machine learning systems to effectively apply prior knowledge to unfamiliar domains and robustly respond to anomalous inputs. The paper argues that despite the success of UniDA in computer vision, its application in language inputs remains under-explored.

**Main Contributions:**

1. **Comprehensive Benchmark for NLP:** The paper presents a comprehensive benchmark for evaluating the generalizability and robustness of NLP models. This benchmark covers multiple datasets with different difficulty levels and characteristics, including temporal shifts and diverse domains, offering a thorough viewpoint of a model's capabilities.

2. **New Metrics:** To better understand the challenges posed by UniDA in the NLP context, the authors propose two novel metrics: Performance Drop Rate (PDR) and Distinction Difficulty Score (DDS). These metrics are designed to quantitatively measure the domain gap and category gap.

3. **Evaluation of UniDA Methods:** The paper provides an empirical evaluation of existing UniDA methods, originally designed for image inputs, within the context of natural language. It also compares these methods with state-of-the-art domain adaptation techniques from NLP literature.

4. **Insightful Findings:** The study reveals that UniDA methods, initially conceived for image inputs, can be effectively transferred to the natural language domain. It also highlights the significant role of adaptation difficulty in determining the performance of these methods.

5. **Testbed for Evaluating UniDA:** A significant contribution of the paper is the introduction of a testbed for evaluating UniDA's performance in the NLP field. This testbed, featuring various levels of domain and category gaps across different datasets, serves as a valuable resource for future research in this area.

**Questions For The Authors:**

A. Can you justify why the H-scores of more challenging tasks, such as MASSIVE, are higher than easier tasks?

**Reasons To Accept:**

**Reasons to Accept:**

1. **Novel Topic Exploration:** The paper is one of the first to explore the applicability of Universal Domain Adaptation (UniDA) in the field of NLP, potentially enhancing the robustness of NLP models.

2. **Comprehensive Benchmark and Testbed:** The provision of a comprehensive benchmark and a robust testbed for evaluating UniDA's performance in NLP is a significant strength of the paper. These resources can accelerate research in this area by providing a well-structured foundation for future work.

3. **Introduction of New Metrics:** The paper presents two new metrics - Performance Drop Rate (PDR) and Distinction Difficulty Score (DDS) - which can be instrumental in assessing domain and category gaps. These metrics add to the paper's novelty and can benefit the wider NLP community.

4. **Empirical Evaluation:** The paper provides a thorough empirical evaluation of existing UniDA methods, extending their use from image to language inputs. It provides valuable insights into how these methods perform in a new context, potentially aiding future research efforts in UniDA for NLP.

5. **Benefit to NLP Community:** The research presented in this paper can contribute to improving the robustness of NLP models against distributional shifts, a significant challenge in deploying machine learning systems in real-world scenarios. This is directly beneficial to the NLP community.

Overall, the paper demonstrates the substantial potential to influence future work in the domain adaptation field of NLP, both by providing a novel framework for understanding the application of UniDA and by offering resources for further exploration and development.

**Reasons To Reject:**

It is a good work that I do not have strong reasons to reject it, and several limitations have been clearly stated by the authors for future work.

1. **Limited Model Size Evaluation:** The study only conducted evaluations with models of limited sizes. While the authors noted that the evaluation of large language models (LLMs) is part of their future work, the absence of their performance assessment limits the comprehensive understanding of UniDA methods across varying model sizes in the NLP context.

2. **Focused on Classification Tasks:** The authors explicitly mention that their proposed testbed is limited to text classification tasks. While this focus may have been practical, it also restricts the generalizability of the findings. Given the diverse range of tasks in NLP, such as named entity recognition, part-of-speech tagging, and sentiment analysis, this limitation might reduce the applicability of the study's contributions.

3. **Lack of Evaluation of Zero-shot and Few-shot Models:** The current trend in NLP includes models with zero-shot and few-shot learning capabilities, but these were not evaluated in the study. Including such models might have provided a more comprehensive evaluation of UniDA's performance.

In conclusion, while the paper provides important contributions, it has several limitations that may affect the comprehensiveness and applicability of its findings. These limitations could be addressed in future work to enhance the paper's impact and relevance to the broader NLP community.

**Reproducibility:**

4: Could mostly reproduce the results, but there may be some variation because of sample variance or minor variations in their interpretation of the protocol or method.

**Reviewer Confidence:**

4: Quite sure. I tried to check the important points carefully. It's unlikely, though conceivable, that I missed something that should affect my ratings.

---

> ### Author Rebuttal · Authors · 2023-08-28
>
> We’d like to express our gratitude for your insightful and positive comments. Here are the responses to the comments.
>
>
> **W1. Limited Model Size**
>
> As recent LLMs are capable of a wide variety of real-world applications, we strongly agree on the importance of exploring the LLM’s robustness from a Universal Domain Adaptation (UniDA) perspective.
> However, in this current work, we have limited the scope to introducing UniDA by proposing a solid benchmark and validating existing state-of-the-art methods within the existing computer vision and NLP literature.
> To ensure the closest alignment with the existing literature in NLP, we conducted all the experiments with the same backbone as the previous domain adaptation studies.
> As the research is still in its early stages, further exploration considering LLMs is a great topic for future work.
>
>
> **W2. Focused on Classification Tasks**
>
> We acknowledge that our work is limited to text classification tasks, and we fully agree that our work can be expanded to diverse tasks such as NER, POS tagging, and even text generation.
> However, the scope of our work was primarily influenced by existing UniDA literature in the vision domain, which is predominantly focused on the classification task.
> As we are the first to introduce the UniDA concept to the NLP field, we considered it appropriate to begin from the same scope as existing works.
> Additionally, the selective task preference is also due to the challenge of defining concepts such as domain and category shifts in tasks such as text generation.
> Despite the weaknesses, we believe our work provides sufficient empirical analysis within the text classification tasks and offers a substantial array of insights.
>
>
> **W3. Lack of Zero-shot and Few-shot Models**
>
> As the majority of models capable of zero-shot and few-shot inference are LLMs, our response remains consistent with W1.
>
>
> **Q1 : Why do some challenging tasks show a higher H-score compared to easier tasks?**
>
> We proposed two different metrics (PDR, DDS) to capture the challenges of UniDA (domain gap and category gap).
> Specifically, PDR quantifies the relative performance degradation, irrespective of the absolute difficulty of the task.
> Let us compare the results of MASSIVE and Huffpost (2016) for a better explanation. (Please refer to Table 1 of the paper and the table below).
>
> For the **MASSIVE** dataset, 98.01 was observed as the source accuracy, indicating that the in-domain task is considerably easy.
> The target accuracy is 62.58, yielding a high PDR value of 36.14, as calculated by Equation 1 of the paper.
>
> On the other hand, the **Huffpost (2016)** dataset is notably difficult, even within the in-domain context.
> Specifically, with a source accuracy of 82.15 and a target accuracy of 61.61, a relatively lower PDR value of 25.00 was observed.
>
> As a result, MASSIVE exhibits a higher PDR value compared to Huffpost (2016), indicating a relatively challenging scenario (i.e., a more significant domain gap).
> However, if we compare the absolute performance in the target domain, MASSIVE reports a higher performance of 62.58, while Huffpost (2016) shows 61.61.
>
> Despite MASSIVE being a challenging task with a high PDR value, it is possible to observe a higher H-score compared to easier tasks such as Huffpost (2016).
>
> |  | Source Accuracy | Target Accuracy | PDR |
> | --- | --- | --- | --- |
> | Huffpost (2016) | 82.15 | 61.61 | 25.00 |
> | MASSIVE | 98.01 | 62.58 | 36.14 |

---

### Official Review · Reviewer_wuhF · 2023-08-04

**Soundness:** 4

**Excitement:**

2: Mediocre: This paper makes marginal contributions (vs non-contemporaneous work), so I would rather not see it in the conference.

**Paper Topic And Main Contributions:**

While UniDA has led to significant progress in computer vision, its application to language input still needs to be explored, despite its feasibility. This paper proposes a comprehensive benchmark for natural language that offers thorough viewpoints on the model’s generalizability and robustness. The benchmark encompasses multiple datasets with varying difficulty levels and characteristics, including temporal shifts and diverse domains. On top of this testbed, we demonstrate that UniDA methods originally designed for image input can be effectively transferred to the natural language domain while also underscoring the effect of adaptation difficulty in determining the model’s performance.

**Reasons To Accept:**

1. The testbed includes datasets divided according to difficulty levels, evaluated by two new evaluation metrics. By dividing the datasets based on the degree of difficulty, the capabilities of each method can be better distinguished.
2. The experiment is relatively complete, with many types of datasets and a comprehensive comparison of different methods. The effectiveness of the UniDA methods can be demonstrated.

**Reasons To Reject:**

The proposed approach simply migrates existing methods from computer vision to solve the generalization problem in the field of natural language processing, limiting the novelty of this paper. Additionally, it is expected that experiments will be conducted to test the generalization ability in more NLP applications, beyond text classification.

**Reproducibility:**

4: Could mostly reproduce the results, but there may be some variation because of sample variance or minor variations in their interpretation of the protocol or method.

**Reviewer Confidence:**

3: Pretty sure, but there's a chance I missed something. Although I have a good feel for this area in general, I did not carefully check the paper's details, e.g., the math, experimental design, or novelty.

---

> ### Author Rebuttal · Authors · 2023-08-28
>
> Thank you for dedicating your valuable time and effort to reviewing our paper. Our responses to the comments are as follows.
>
>
>
> **W1. Limited Novelty**
>
> The reviewer has pointed out that our work is a simple migration of methods from the vision domain,
> but we would like to strongly argue that the main contribution of our work is to address the importance of relatively unexplored Universal Domain Adaptation (UniDA) in the NLP field and serve as the cornerstone of pioneering and setting a solid environment for future research.
> Specifically, we made efforts to construct a benchmark including various types of shifts in NLP (e.g., sentiment, temporal shifts) and different difficulty levels. (Section 3.3)
> The levels of difficulty were quantitatively measured by two novel metrics (PDR and DDS) which measures the degree of domain and category gap. (Section 3.1)
>
> As the reviewer pointed out, existing methods from the computer vision domain were migrated and evaluated on our benchmark.
> However, we also made efforts to evaluate state-of-the-art methods from the NLP domain for comparative analysis between existing methods.
> By verifying the effectiveness of prior studies within this benchmark, we believe we have laid a robust foundation for future UniDA research.
> We humbly urge that our contribution involves efforts beyond simple migration, aiming to integrate and solidify the UniDA setting in the NLP field.
> Your thoughtful consideration of this aspect would be greatly appreciated.
>
>
>
> **W2. Limited to Text Classification**
>
> We agree with the reviewer’s point and are fully aware that our work is concentrated only on text classification tasks.
> The fact that existing UniDA literature in the vision domain is predominantly focused on the classification task has significantly influenced the scope of our research.
> As our work is the first to introduce the concept of UniDA to the NLP field, we deemed it appropriate to follow existing works and begin the research within the classification tasks.
> As mentioned in the Limitation Section, the selective task preference is also due to the challenge of defining concepts such as domain and category shifts in tasks such as text generation.
> We advocate for gradually expanding UniDA’s scope to encompass advanced tasks such as text generation.
> Despite the limitation, we believe our paper provides comprehensive empirical analysis exclusively within the text classification task, offering diverse insights.

---

### Meta-Review · Area_Chair_QM9K · 2023-09-16

**Recommendation:** 3

**Metareview:**

This paper looks at the task of generalization and  domain adaptation borrowing ideas from previous work (Universal Domain Adaptation) from computer vision, and applying them to NLP datasets. The paper presents benchmark tasks, alternate metrics, and empirically evaluate the ideas of Universal Domain Adaptation on the benchmarks and find them to transfer well.

The reviewers have rated the paper strong on soundness but excitement scores vary. The authors address some of the concerns of reviewers in terms of limited baselines, and In particular reviewers have noted several limitations of the work: namely that it is confined to classification tasks, limited model size/architecture, and is missing some zero-shot baselines from large language models which would have been appropriate for this topic. Nevertheless the experiments are mostly sound even if missing interesting comparisons.

---

### Decision · Program_Chairs · 2023-10-07

**Decision:**

Accept-Findings

**Comment:**

This paper looks at the task of generalization and  domain adaptation borrowing ideas from previous work (Universal Domain Adaptation) from computer vision, and applying them to NLP datasets. The paper presents benchmark tasks, alternate metrics, and empirically evaluate the ideas of Universal Domain Adaptation on the benchmarks and find them to transfer well.

The reviewers have rated the paper strong on soundness but excitement scores vary. The authors address some of the concerns of reviewers in terms of limited baselines, and In particular reviewers have noted several limitations of the work: namely that it is confined to classification tasks, limited model size/architecture, and is missing some zero-shot baselines from large language models which would have been appropriate for this topic. Nevertheless the experiments are mostly sound even if missing interesting comparisons.